# CC-DTW: An Accurate Indoor Fingerprinting Localization Using Calibrated Channel State Information and Modified Dynamic Time Warping

**DOI:** 10.3390/s19091984

**Published:** 2019-04-28

**Authors:** Zhongliang Deng, Xiao Fu, Qianqian Cheng, Lingjie Shi, Wen Liu

**Affiliations:** School of Electronic Engineering, Beijing University of Posts and Telecommunications, Beijing 100876, China; dengzhl@bupt.edu.cn (Z.D.); ambercheng@bupt.edu.cn (Q.C.); shilj@bupt.edu.cn (L.S.); liuwen@bupt.edu.cn (W.L.)

**Keywords:** channel state information, dynamic time warping, spatial resolution, indoor positioning, time reversal

## Abstract

Indoor wireless local area network (WLAN) based positioning technologies have boomed recently because of the huge demands of indoor location-based services (ILBS) and the wide deployment of commercial Wi-Fi devices. Channel state information (CSI) extracted from Wi-Fi signals could be calibrated and utilized as a fine-grained positioning feature for indoor fingerprinting localization. One of the main factors that would restrict the positioning accuracy of fingerprinting systems is the spatial resolution of fingerprints (SRF). This paper mainly focuses on the improvement of SRF for indoor CSI-based positioning and a calibrated CSI feature (CCF) with high SRF is established based on the preprocess of both measured amplitude and phase. In addition, a similarity calculation metric for the proposed CCF is designed based on modified dynamic time warping (MDTW). An indoor fingerprinting method based on CCF and MDTW, named CC-DTW, is then proposed to improve the positioning accuracy in indoors. Experiments are conducted in two indoor office testbeds, and the performances of the proposed CC-DTW, one time-reversal (TR) based approach and one Euclidean distance (ED) based approach are evaluated and discussed. The results show that the SRF of CC-DTW outperforms the TR-based one and the ED-based one in both two testbeds in terms of the receiver operating characteristic (ROC) curve metric, and the area under curve (AUC) metric.

## 1. Introduction

With an increase in the demand for ubiquitous location-based services (LBS), localization and navigation applications have become more important in daily life [1]. Considering the fact that people today spend more than 80% of their time in indoor environments [2], indoor location-based services have gained considerable attention recently, with a market value predicted to reach US $10 billion by 2020 [3]. Although widely used global navigation satellite systems (GNSS) could provide a relatively high positioning accuracy outdoors, it does not perform well indoors due to the coverage limitation of satellite signals. Therefore, various indoor positioning technologies have been proposed in order to enable high-accuracy localization in indoors over the past two decades, including infrared [4], ultrasonic [5], ultra-wideband (UWB) [6], pseudolite [7], acoustic signals [8], Wi-Fi [9,10], Bluetooth [11], etc. However, indoor high-accuracy positioning is still an open but challenging issue, which has been proved by the conclusion “the indoor location problem is not solved” from the recent indoor localization competitions hosted by Microsoft [12].

Among all the technologies employed for indoor positioning, Wi-Fi is still considered one of the most promising schemes due to its ubiquity and flexible deployment. The received signal strength (RSS) and the channel state information (CSI) are two essential positioning features used in indoor Wi-Fi localization systems. The RSS is widely used in Wi-Fi fingerprinting localization, which can be divided into two stages: the offline training stage and online positioning stage. In the offline stage, the location-dependent RSS is collected at certain positions and a radio map based on the measured RSS is then constructed for online matching. The classical RSS-based Wi-Fi fingerprinting systems include RADAR [13] and Horus [14], which have been reported to achieve an average accuracy of 3–5 m and 2 m, respectively. Recent works utilize the information from other sensors as the assistance to RSS-based fingerprinting systems. The work in [15] proposes an RSS-based positioning method with the Li-Fi assisted coefficient calibration and has achieved an improvement in positioning accuracy. A fingerprint fusion method based on weighted least squares, which uses both the RSS in Wi-Fi based system and the information in inertial navigation system, is proposed in [16], and an average localization accuracy of 2.03 m has been achieved. Although RSS-based positioning systems could provide meter-level accuracy, there are still some weaknesses when considering RSS-based systems in dense cluttered indoor environments. For example, the RSS varies with time at a fixed position due to the multipath effects [14] and the RSS is a coarse measurement which lacks the frequency information to capture the multipath property [17]. Therefore, the RSS does not have enough spatial resolution, which in other words could not describe the wireless signal’s characteristics in detail or with enough accuracy, and would introduce undesirable errors into fingerprinting systems.

The CSI, which can be obtained in Wi-Fi OFDM systems to provide fine-grained channel response estimations, is able to discriminate multipath characteristics and thus holds the potential for the convergence of accurate and pervasive indoor localization [18]. The primary advantage of CSI over RSS is that CSI could estimate the wireless signal transmission channel on each subcarrier in the frequency domain, which increase the dimension of the positioning feature and takes the multipath information into consideration. Meanwhile, the CSI could stay fairly stable at same position over time [19]. These advantages have made CSI a desirable candidate feature for indoor high accuracy positioning.

The CSI-based indoor Wi-Fi positioning systems can be divided into range-based positioning and fingerprinting. The range-based positioning technologies mainly measure the distance, time of flight (TOF), angle of arrival (AOA) or other measurements between transmitter and receiver, and perform trilateration or triangulation to estimate the receiver’s position. FILA [20], which utilized the effective CSI amplitude and the relationship between the effective CSI and the distance to perform ranging, was proposed to alleviate multipath effect and has achieved sub-meter level positioning accuracy in several indoor environments. Chronos [21] obtained a sub-nanosecond TOF by measuring discontinuous frequency bands based on the inverse non-uniform discrete Fourier transform. PILA [1] was proposed to estimate the AOA from the measured CSI, and 0.7 m positioning accuracy has been achieved through two-dimensional spatial smoothing. SpotFi [22] estimated the AOA and TOF jointly based on the rebuilt CSI matrix and eigenvalue decomposition, and has achieved a sub-meter level localization. The range-based CSI positioning could achieve a relatively high positioning accuracy. However, the distance estimation needs empirical models, and the measurement of TOF or AOA usually needs additional equipment.

The CSI-based fingerprinting technologies regard location-specific features as fingerprints and perform matching or machine learning methods to estimate positions. Different features and different feature similarity calculation metric are proposed recently in order to improve the positioning accuracy of CSI-based fingerprinting. FIFS [17] utilized the CSI amplitude values as fingerprints and adopted a probabilistic model to determine the target location. PinLoc [23] also utilized the CSI amplitude extracted from off-the-shelf Intel WiFi Link 5300 Network Interface Card (IWL 5300 NIC) as fingerprints, while considering $1\times 1\;m^{2}$ spots for training data. PhaseFi [24] proposed a CSI phase calibration method, utilized the calibrated phase information as fingerprints and has achieved meter-level accuracy based on deep learning. CSI-MIMO [25] used the differences in amplitude and phase between adjacent subcarriers as fingerprints and has achieved 0.95 m accuracy in lab environments. DeepFi [26] collected CSI amplitude values to train the deep belief networks and the weights of the deep network were treated as fingerprints. BiLoc [27] utilized the CSI amplitude and the estimated AOA as fingerprints, and Amp-Phi [28] exploited the amplitude and phase information at the same time to establish a fingerprinting database. HATRFLA [29] proposed a location-specified fingerprint based on the product of the time reversal resonating strength (TRRS) from both CSI amplitude and phase, and has achieved centimeter-level positioning accuracy in an area of approximately 50 cm×50 cm. The work in [30] utilized both the RSS and CSI as fingerprints and established four deep neural networks with a one-dimensional convolutional neural network to form new positioning features for indoor localization. The CSI-based fingerprinting needs no extra equipment, and the Spatial Resolution of the Fingerprints (SRF) is one of the main factors that could influence the positioning accuracy.

The SRF can be regarded as the area of an ambiguous region, in which the target location and its nearby locations cannot be distinguished accurately according to the fingerprints’ similarity comparison. The higher the SRF is, the smaller the area is, which means that the fingerprint together with its similarity calculation metric are more suitable for fine-grained fingerprinting systems. There exists two ways to improve the SRF, one is to explore new types of fingerprints, and the other is to establish new similarity calculation metric. Using both amplitude and phase information at multiple subcarriers to form new positioning features would have the potential to improve the SRF, like the previous works done in [25,27,28,29]. Exploring a new similarity calculation metric for fingerprints could also improve the SRF. The widely used metric is the Euclidean distance (ED) based metric, which evaluates the similarity of two fingerprints based on the Euclidean distance. Time reversal (TR) based method was proposed recently to calculate the similarity between two fingerprints and has been proven more accurate and robust in multipath environments [31,32]. The work in [31] utilized the TRRS as the similarity calculation metric and has achieved a 5 cm accuracy through bandwidth concatenation in an area of 20 cm×70 cm environment. The work in [32] has shown that through diversity exploitation in frequency and spatial domain, the TRRS metric could realize centimeter-level accuracy. TRRS metric is a promising candidate to improve the SRF for indoor high accuracy positioning, however, the performance of TRRS would deteriorate with limited bandwidth.

This paper mainly focuses on the improvement of SRF in two aspects: new type of fingerprint and new similarity calculation metric. Further, an indoor CSI-based fingerprinting method named CC-DTW is proposed to improve the SRF as well as the positioning accuracy. The main contributions of this paper are as follows:A new type of fingerprint named Calibrated CSI Feature (CCF) is proposed with the aim of improving the SRF. The CSI amplitude is denoised based on the mode value, and the CSI phase is denoised based on linearization after unwarping. Both the processed amplitude and phase information are integrated based on the formation of original CSI, which is called CCF, in order to reduce computational complexity in feature matching.A new similarity calculation metric based on modified dynamic time warping (MDTW) is established to compute the similarity between the proposed CCF. To our knowledge, this is the first time that the DTW method has been introduced into fingerprints’ similarity calculation.A fine-grained fingerprinting method based on CCF fingerprint and MDTW metric, named CC-DTW is then proposed and implemented in a mainstream 2.4 GHz Wi-Fi system with 20 MHz bandwidth and three receiving antennas in two indoor office environments. The performance of CC-DTW is evaluated compared with one TR-based approach and one ED-based approach.

The remainder of this paper is arranged as follows: Section 2 presents the preliminaries and establishes the CCF; Section 3 studies the MDTW metric and designs the CC-DTW method; the experiments and results are presented in Section 4; finally, Section 5 concludes our work.

## 2. Preliminaries and the Calibrated CSI Feature

This section firstly presents some preliminaries about CSI amplitude and phase information. Then the preprocessing method for both CSI amplitude and phase are analyzed, followed by the formation of CCF.

### 2.1. Preliminaries

#### 2.1.1. CSI Introduction

The wireless signals travel through multiple paths from transmitters to receivers in indoor environments, which is referred as multipath effects. The time delay, amplitude decay and the phase shift are different at each path. The wireless channel is usually modeled as a spatial linear filter [33], which can be described using channel impulse response (CIR). The CIR of one wireless channel can be formulated as Equation (1).
(1)$$h\left(\tau \right)=\sum_{i=1}^{N}\alpha_{i}e^{-j\theta_{i}}\delta \left(\tau -\tau_{i}\right),$$
where h(τ) stands for the CIR, and *N* is the number of transmission paths. $\alpha_{i}$, $\theta_{i}$, and $\tau_{i}$ are amplitude, phase and delay for the *i*th path, respectively. By applying fast Fourier transformation (FFT) to the CIR, the channel frequency response (CFR) can be derived as in Equation (2).
(2)$$H\left(\omega \right)=FFT\left[h\left(\tau \right)\right]=\sum_{i=1}^{N}\alpha_{i}e^{-j\left(\omega \tau_{i}+\theta_{i}\right)},$$
where H(ω) stands for the CFR, and $\alpha_{i}$, $\theta_{i}$ also stand for amplitude and phase, respectively.

In the orthogonal frequency division multiplexing (OFDM) and multiple in multiple out (MIMO) systems, the CFR can be sampled at predetermined intervals and represented by the amplitude and phase values at multiple subcarriers, which introduces the definition of CSI data. The CSI data collected in a Wi-Fi OFDM system can be given as follows.
(3)$$H\left(f\right)=\left[H\left(f_{1}\right),H\left(f_{2}\right),\ldots ,H\left(f_{k}\right)\right],$$
where H(f) stands for the CSI data and *k* is the number of subcarriers. $H\left(f_{k}\right)$ is the CSI data for subcarrier wave $f_{k}$, which is combined with amplitude and phase information. The expression of $H\left(f_{k}\right)$ is given in Equation (4).
(4)$$H\left(f_{k}\right)=\left|H\left(f_{k}\right)\right|e^{j∠H\left(f_{k}\right)},$$
where $\left|H\left(f_{k}\right)\right|$ and $∠H\left(f_{k}\right)$ are CSI amplitude information and phase information respectively.

CSI is the information that represents the channel properties of a transmission link and describes how the wireless signals propagate from a transmitter to a receiver. The CSI data contains the effects of fading, power decay and scattering with distance, hence it can be utilized for indoor positioning.

#### 2.1.2. CSI Used in This Paper

This paper mainly focuses on indoor Wi-Fi fingerprinting, and thanks to the IEEE 802.11n protocol [34] and Intel, the CSI data can be obtained and extracted using the commodity Wi-Fi devices and IWL 5300 NIC. In consideration of the limited bandwidth of commonly used devices and the flexibility of system deployment, this paper only utilizes a single 20MHz channel at 2.4 GHz Wi-Fi frequency band together with three receiving antennas and one transmitting antenna. The experimental settings are simple, but accurate localization under such a situation is still a challenge. The measured CSI data is formed into a CSI matrix with the size of P×A×K for processing and positioning, where *P*, *A* and *K* are the number of data packages, antennas and subcarriers, respectively. Specifically, *A* equals to 3 and *K* equals to 30 in this paper.

### 2.2. CSI Amplitude Preprocessing

The CSI amplitude information is location-specific because the multipath conditions differ at positions, so as the amplitude of paths. In order to intuitively show the location-specific characteristic of CSI amplitude, we measured the CSI data at two test positions with an interval of 1.6m and calculated the amplitude of each package for all subcarriers and antennas. Figure 1 shows the measured CSI amplitude values at these two positions extracted from the same antenna in about 1000 packages. From Figure 1a,b it can be concluded that CSI amplitude has an obvious difference between different positions. Figure 1 also shows that the raw measured CSI amplitude usually has noises and cannot be used for localization without preprocessing.

In order to preprocess the CSI amplitude, the boxplot diagram is firstly utilized to analyze the outliers of amplitude values, which is shown in Figure 2. It can be concluded that the amplitude at same subcarrier is nearly stable at different packages except for the influence of outliers, since the range between the upper quartile and the lower quartile is small, for example 0.43 dB for position 1 and 0.85 dB for position 2. This motivates us to propose a mode value-based amplitude denoising and feature extracting method, which could reduce the computational complexity at the same time.

The mode value of the CSI amplitude stands for the value which has the largest occurrence number in the whole packages for one subcarrier. Since the CSI amplitude for each subcarrier is stable enough as shown in Figure 2, extracting the mode value and utilizing it as the feature of CSI amplitude for the corresponding subcarrier is noise-insensitive and feasible. Meanwhile, the computational complexity of mode value calculation is lower than clustering-based approaches used in [29,35], which makes the mode value an efficient and lightweight candidate for CSI amplitude fingerprint. The proposed mode value of CSI amplitude can be formulated by Equation (5).
(5)$$Mode\left(f_{k}\right)=\underset{\eta }{\max}\left\{\hat{A}mp\left(f_{k,u}\right):\eta =occurrence\left(\hat{A}mp\left(f_{k,u}\right)\right)\right\},$$
where $Mode\left(f_{k}\right)$ stands for the amplitude mode value of *k*th subcarrier, and η stands for the occurrence number of measured amplitude values. $\hat{A}mp\left(f_{k,u}\right)$ stands for the *u*th unique value of the measured amplitude for *k*th subcarrier, and *u* is the number of unique values for *k*th subcarrier.

After mode value extraction, the amplitude mode values of the whole 30 subcarriers at one position are formed into one fingerprint, called the calibrated amplitude. The general expression of the calibrated amplitude is given in Equation (6).
(6)$$Amp_{C}\left(f_{k}\right)=Mode\left(f_{k}\right),$$

The calibrated amplitude based on mode value of the above two test positions are shown in Figure 3. It can be concluded that the calibrated amplitude remains the trend of original amplitude and the location-specific characteristic and removes the noise in the same time.

### 2.3. CSI Phase Preprocessing

The measured CSI phase information could also not be used for indoor localization directly. Except for the measured noise, the measured phase is folded due to the recurrence characteristic of phase [24]. In order to show this characteristic, the measured phase values at the same two positions as in Section 2.2 of three receiving antennas are plotted in Figure 4. It is noticeable that the measured phase is folded with the increase of subcarrier index and the phase ranges from −π to π. In order to analyze the true phase changing trend and turn the measured phase into one positioning feature, we firstly unwrap the measured phase. The unwrapped phase values for these two positions are given in Figure 5.

It is noticeable in Figure 5 that the unwrapped phase values during all the subcarriers have the same changing trend, but the phase values decrease with the subcarrier index. The reason is that the carrier frequency offset (CFO) and the sampling frequency offset (SFO) exist in phase information. In general, the unwrapped phase can be expressed as follows [23].
(7)$${\hat{\varphi }}_{k}=\varphi_{k}-2\pi \frac{l_{k}}{M}\delta +\beta +Ζ,\;k\in \left[1,K\right],\;K=30,$$
where $\varphi_{k}$ is the true phase and ${\hat{\varphi }}_{k}$ is the measured phase; δ is the time lag due to SFO, and β is the unknown phase offset due to CFO; $l_{k}$ is the subcarrier index of *k*th subcarrier, M is the FFT size, and Z is the measurement noise. The value of δ and β is hard to evaluated, so we adopt a linear transformation [24] to mitigate the effects of these two errors approximately. The slope and offset value for phase linear calibration used in this paper are given as follows.
(8)$$k=\frac{{\hat{\varphi }}_{K}-{\hat{\varphi }}_{1}}{l_{K}-l_{1}},$$
(9)$$b=\frac{1}{l_{K}}\sum_{k=1}^{K}{\hat{\varphi }}_{k},$$
where *k* stands for the slope and *b* stands for the offset of calibration. ${\hat{\varphi }}_{k}$ and $l_{k}$ have the same definition as the one in Equation (7). The calibrated phase through the linear calibration above can be expressed as follows.
(10)$${\tilde{\varphi }}_{k}={\hat{\varphi }}_{k}-kl_{k}-b,$$
where ${\tilde{\varphi }}_{k}$ is the phase value after calibration. The calibrated phase values of at the two test positions are shown in Figure 6. It can be concluded that after unwarping and linearization, the calibrated phase information is unfolded and remains the location-specific characteristics, which can be utilized as fingerprints. The general expression of the calibrated phase is given in Equation (11), which would be integrated with calibrated amplitude to form the proposed CCF.
(11)$$Pha_{C}\left(f_{k}\right)={\tilde{\varphi }}_{k},$$

### 2.4. Formation of CCF

After the preprocess of CSI amplitude and phase in Section 2.2 and Section 2.3, the location-specific positioning features, called the calibrated amplitude and the calibrated phase, are obtained. Instead of directly use these two features to construct the positioning fingerprint, this paper proposes to integrate these features into one fingerprint, called CCF. The CCF is formulated in Equation (12).
(12)$$CCF\left(f_{k}\right)=Amp_{C}\left(f_{k}\right)e^{-jPha_{C}\left(f_{k}\right)},$$
where $CCF\left(f_{k}\right)$ is the proposed positioning feature, $Amp_{C}\left(f_{k}\right)$ and $Pha_{C}\left(f_{k}\right)$ are the calibrated amplitude and phase, respectively.

There are several reasons for this integration of CCF. One is that through the amplitude and phase integration, both the two positioning features could be used in location estimation at the same time. Compared with the methods using only one feature, the CCF introduces multi-dimensions for the fingerprints, which could improve the SRF and the positioning accuracy. However, higher dimensions always mean larger computational complexity in fingerprint matching. Suppose that there are *N* dimensions for each fingerprint, such as amplitude and phase. For each fingerprint matching procedure, the computational complexity of the methods using amplitude and phase separately is about *O*(2 × *N*). The CCF could perform the matching of amplitude and phase through one calculation because of the integration, and the computational complexity for CCF is about *O*(*N*). In other words, the CCF could reduce the complexity of fingerprint matching when comparing with the methods using amplitude and phase separately. The last, but not least reason is that this integration tries to remain the original expression of channel information as far as possible, which could ensure the feasibility of this integration. The performance of the CCF in the SRF improvement will be evaluated in Section 4.

## 3. CC-DTW: Fine-Grained Indoor Fingerprinting Based on CCF and MDTW

This section firstly gives an overview of the fine-grained indoor fingerprinting method named CC-DTW based on the formation of CCF. A new similarity calculation metric named MDTW is then proposed, which is used in CC-DTW to compute the similarity between CCF fingerprints. Finally, two evaluation indicators are introduced in order to evaluate the SRF improvement performance of the CC-DTW.

### 3.1. Overview of CC-DTW

The flowchart of the proposed CC-DTW fingerprinting method is shown in Figure 7. Like other indoor fingerprinting methods, the CC-DTW consists of two stages: the offline training stage and the online matching stage. As shown in Figure 7, in the offline training stage, the CSI data is collected at certain positions and both the amplitude and phase data are preprocessed using the method in Section 2. Then the calibrated amplitude and phase are integrated in the form of CCF for each position and stored as fingerprints in the database. In the online matching stage, the CSI data is measured at unknown positions, and the CSI phase is preprocessed using the unwarping and linearization method proposed in Section 2.3. After phase calibration, the measured CSI is transformed into measured CCF according to the formation in Section 2.4, and the similarity calculation metric based on MDTW is applied to compute the similarity between measured CCF and the CCF in database in order to perform position estimation. The similarity calculation metric used in CC-DTW is explained in detail in the following subsections.

### 3.2. The MDTW Similarity Calculation Metric

The similarity calculation metric is also one of the factors that can influence the SRF as well as the positioning accuracy of indoor fingerprinting systems. The generally used metric is based on the Euclidean distance. A TR-based similarity calculation metric has recently been proposed and utilized to compute the similarity between two fingerprints, which has been proven to outperform its traditional Euclidean distance-based counterparts in [29,31,32]. However, the performance of TR-based approaches would deteriorate with the limited channel bandwidth. In order to propose an accurate as well as suitable for limited bandwidth similarity calculation metric, this paper established the MDTW metric inspired by the dynamic programming and the dynamic time warping algorithm.

The DTW algorithm is widely used in speech signal recognition and matching. It can also be used to calculate the similarity between two features [35]. Based on the dynamic programming, the DTW algorithm could find the minimum warping path between two sequences, and the cumulated distance of the warping path can be used as an indicator to evaluate the similarity. The DTW algorithm can be formulated as Equation (13).
(13)$$DTW\left(Q,T\right)=\min\left\{\sqrt{\sum_{r=1}^{R}\varpi_{r}}/R\right\},$$
where DTW(Q,T) is the dynamic time warping path between two sequences Q and T. R is the length of the warping path, which can be used to compensate the difference between warping paths. $\varpi_{r}$ is the *r*th node in the path, which can be expressed as $\varpi_{r}={\left(i,j\right)}_{r}$, and i, j stands for the *i*th and *j*th element in sequences Q and T, respectively.

This paper utilizes the cumulated distance of two fingerprints’ warping path obtained from DTW as the reference of the similarity of these fingerprints. Considering that the fingerprints used in CC-DTW are complex values, which consists amplitude and phase, this paper modified the distance matrix in DTW as the 2-norm of the difference vector between two CCF fingerprints, instead of the subtraction. The general expression of the modified distance matrix of two CCF fingerprints is shown as follows.
(14)$$d\left(Q_{i},T_{j}\right)={\left\|CCF_{Q}\left(f_{i}\right)-CCF_{T}\left(f_{j}\right)\right\|}_{2},$$
where $d\left(Q_{i},T_{j}\right)$ stands for the modified distance of two CCF fingerprints Q and T. $Q_{i}$ and $T_{j}$ are the *i*th and *j*th elements in the two CCF fingerprints, which can be expressed as $CCF_{Q}\left(f_{i}\right)$ and $CCF_{T}\left(f_{j}\right)$, respectively.

Based on the modified distance matrix, the cumulated distance of the minimum warping path of two CCF fingerprints using MDTW algorithm can be expressed as in Equation (15).
(15)$$\gamma \left(i,j\right)=d\left(Q_{i},T_{j}\right)+\min\left\{\gamma \left(i-1,j-1\right),\gamma \left(i-1,j\right),\gamma \left(i,j-1\right)\right\},$$
where γ(i,j) stands for the cumulated distance from (1,1) node to (i,j) node in the warping path. $d\left(Q_{i},T_{j}\right)$ is the modified distance, which can be calculated according to Equation (14). Since each CCF fingerprint has 30 subcarriers, the cumulated distance between two CCF fingerprints can be derived as γ(K,K), where K equals to 30.

Since the cumulated distance stands for the shortest distance between two CCF fingerprints based on dynamic programming, the similarity between CCF fingerprints should be inversely proportional to the cumulated distance. Using the derived cumulated distance between two CCF fingerprints with MDTW algorithm, the similarity of these two fingerprints can be calculated as follows.
(16)μQ,T=1[γ(K,K)+ε],
where $\mu_{Q,T}$ is the similarity between CCF fingerprint Q and T based on MDTW. γ(K,K) is the cumulated distance derived in Equation (15), and ε is an infinitesimal in case that the denominator is zero. The CC-DTW utilizes this similarity calculation metric for fingerprint matching and location determination.

### 3.3. SRF Evaluation Indicators

This paper proposes two approaches to improve the SRF as well as the positioning accuracy for indoor fingerprinting systems. One is to form a new fingerprint feature named CCF, and the other is to establish a new fingerprint similarity calculation metric named MDTW. In order to evaluate the performance of the proposed methods in the improvement of SRF, two evaluation indicators are introduced and utilized.

The receiver operating characteristic (ROC) curve and the area under curve (AUC) value are two indicators that can evaluate the performance of classifiers, which has been widely used in machine learning. It can also be introduced into fingerprinting systems to evaluate the similarity of two fingerprints [32]. The ROC curve represents the relationship between the false positive rate (FPR) and the true positive rate (TPR), and the closer the curve is to the TPR-axis, the larger the SRF is. The AUC value is the integral area of ROC curve, and the larger the AUC is, the larger the SRF is. The ROC and AUC indicators can be expressed as follows.
(17)$$ROC=\left(FPR,TPR\right)=\left(\frac{FP}{FP+TN},\frac{TP}{TP+FN}\right),$$
(18)$$AUC=\int_{0}^{1}ROC,$$
where FP, TN, TP, FN stands for the “False Positive”, “True Negative”, “True Positive” and “False Negative”, respectively.

This paper utilizes the ROC and AUC to evaluate the performance of the proposed CCF and MDTW metric, and the experimental results are shown in Section 4.

## 4. Experiments and Results

This section shows the experiments and results. The experimental scenarios together with the system implementation are introduced firstly. The performance of the proposed CCF and MDTW metric in the SRF improvement is then evaluated and discussed, and finally the positioning performance of CC-DTW method is presented and compared with one TR-based approach and one ED-based approach.

### 4.1. Experimental Scenario and Implementation

Experiments are conducted in two real indoor environments in our laboratory. One is in the Room 523 at Main Building of Beijing University of Posts and Telecommunications (BUPT), named Test-bed 1, and the other is in our laboratory at the 9th floor of BUPT Research Building, named Test-bed 2. The Test-bed 1 is with an area of 8 m by 4 m, which consists of tables, chairs, file cabinets and exhibition cabinets. One transmitter as fixed access point and one receiver as mobile device are deployed in Test-bed 1 and 20 test points with an interval of 0.8 m are designed for the experiments. Figure 8 shows the real environment and the planer graph of the Test-bed 1. The red five-pointed star in Figure 8b indicates the position of transmitter, and the red dots indicate the test points.

In order to evaluate the influence of multiple transceivers, as well as the robustness of the proposed methods against different environments, we conducted experiments in Test-bed 2. The Test-bed 2 is with an area of 16.46 m by 8.4 m, which consists of an office room and a meeting room separated by a glass wall. There are plenty of tables and chairs, together with exhibition cabinets and servers in Test-bed 2, which can be regarded as a complex environment with multi-room layout for indoor localization. Due to the hardware limitation, we deployed four receivers as fixed access points and one transmitter as the mobile device that needs to be located, which is contrary to the deployment in Test-bed 1. Fifty-nine test points with an interval of 1.2 m is selected in the public areas in Test-bed 2 for SRF improvement evaluation and positioning test. The experimental environment and the planer graph of Test-bed 2 is shown in Figure 9. The red five-pointed stars in Figure 9 stand for the four receivers in Test-bed 2 and the red dots stand for the test points.

Considering the stability of Wi-Fi signals as well as the flexibility of deployment, all the transceivers used in this paper are industrial personal computers (IPCs) equipped with IWL 5300 NICs, instead of wireless routers and laptops. It should be noted that since the Wi-Fi CSI data cannot be analyzed using most of the commodity mobile phones currently, the IPC is a potential candidate for CSI data measuring and processing. Comparing with laptops, the IPC is small and lightweight, which can also be regarded as a mobile device in indoor localization. The transceivers used in this paper is shown in Figure 10. One antenna is configured for each transmitter and three antennas are configured for each receiver in both the test-beds in our experiments. The transmitter in Test-bed 1 is placed on the top of one worktable at a height about 2 m, and the receiver is carried by a robot at a height about 1.5 m, as shown in Figure 8. The robot used in Test-bed 1 is for the measurements of fingerprints. The receivers in Test-bed 2 are fixed on the wall at a height about 2.4 m, and the transmitter is carried by one of our researchers at a height about 1.2 m.

With the help of Linux CSI tool [36], we modified the NIC’s device driver to read the recorded CSI values over 30 subcarriers for each packet reception. We utilized the 2.4GHz Wi-Fi OFDM signals at a bandwidth of 20 MHz to measure the CSI data. The CSI data is measured at all the test points in the two test-beds twice, one for offline training and the other for testing. We measured CSI data at a sampling rate of 100 packages per second, and at each test point, the CSI data was recorded for about one minute. That is 6000×3×30 CSI data packages for each position. It should be noticed that during the measurements of CSI data, the experimental environments had little changes and the transmitters were kept working. We find that the restart of transmitters could have a huge influence on the stability of CSI data at the same position. In order to evaluate the time stability of Wi-Fi CSI data and the distribution of measurements over time, experiments are conducted in Test-bed 1. Without loss of generality, we select two points in Test-bed 1 and measure the CSI data continuously for three days, with a sampling rate of 30 packages per minute. The CSI data is measured at one of the two points without restart of the transmitter and at the other point with several restarts at the same time. During the measuring stage, the environment remains little change. Both the amplitude and phase are measured and preprocessed using the method in Section 2. Without loss of generality, we select the measurements from one antenna to show the time stability of CSI data and the results are shown in Figure 11.

Figure 11a,b show the measurements of amplitude and phase over time without restart of the transmitter, and Figure 11c,d show the measurements with several restarts. The sudden changes in Figure 11c,d indicate the restarts. It can be concluded that the CSI data remains the same trend at the same point over a relatively long time, which indicates that the CSI data could be time stable under the circumstance that the environments remain little change and the transmitters keep working without restart. Therefore, we keep the transmitters working during the experiments in this paper, and it is also easy to achieve even using the commodity Wi-Fi devices.

### 4.2. SRF Improvement Evaluation and Analysis

In order to evaluate the SRF improvement performance of the proposed method, all the measured CSI data of the training measurement and the testing measurement are preprocessed according to the method in Section 2. The amplitude information and phase information are extracted from the CSI data, calibrated and stored in separated fingerprint database. Meanwhile, the CCF is also formed based on the calibrated amplitude and phase, and stored in another database. In order to describe the databases concise and explicit, we called the amplitude, phase and CCF database extracted from the training measurement “Amp_train”, “Pha_train” and “CCF_train”, respectively. The similar names for the amplitude, phase and CCF database from the testing measurement are “Amp_test”, “Pha_test”, and “CCF_test”. These database names would be used for the following analysis.

This paper evaluates the SRF improvement of the proposed method in three aspects: the aspect of fingerprints which means the improvement of CCF over amplitude and phase features, the aspect of similarity calculation metric which means the improvement of MDTW over TR-based and ED-based approach, and the aspect of system which means the improvement of CC-DTW.

#### 4.2.1. SRF Improvement of the CCF

In this subsection, the SRF improvement is evaluated and analyzed in the aspect of fingerprints. We select the amplitude, phase and CCF as three positioning features and calculate the similarity between each training database and testing database of the two test-beds, such as the “Amp_train” and the “Amp_test”, using the same similarity calculation metric, respectively. Afterwards, the similarity values of these three features are presented and compared, according to the ROC curve and AUC value mentioned in Section 3. Without loss of generality, the MDTW similarity calculation metric is used for all of the three features. The similarities of fingerprints are shown in the form of color map for two test-beds in Figure 12, and the ROC curves together with the AUC values are shown in Figure 13 and Table 1.

Figure 12 shows the similarity of the three features in the form of color map for two test-beds. In Figure 12, the color of each grid indicates the similarity of each pair positions in training database and testing database, and the closer the grid’s color is to red, the closer the similarity is, which in other words is that the corresponding two positions are harder to distinguish using the corresponding feature. Therefore, it can be concluded from Figure 12 that the similarity of CCF has the least red girds in two test-beds, which means the SRF of CCF is better than the others. It can also be concluded from Figure 12 that for the same feature, the SRF in Test-bed 2 is better than that in Test-bed 1, take Figure 12e,f for example. The reason is that in Test-bed 2, there are four receivers used for fingerprinting, which means the increase of feature dimensions in Test-bed 2. However, high dimensions of fingerprints also increase the computational complexity of positioning, which should be balanced when considering indoor localization.

Figure 13 and Table 1 shows the ROC curve and the AUC values of the three features, which can evaluate the SRF quantitatively. It can be concluded from Figure 13a,b that the ROC curve of CCF is the closest to the TPR-axis, which indicates that the CCF has the best SRF for both two test-beds. For a fixed FPR as 0.2, the TPR is increased from 0.95 with amplitude feature to 1 with CCF, and from 0.8 with phase feature to 1 with CCF in Test-bed 1, which is 0.9746 to 0.9831 and 0.9492 to 0.9831 in Test-bed 2. The results show that the CCF has about 5.3% and 25% improvement of SRF in ROC when compared with amplitude feature and phase feature in Test-bed 1, and about 0.8% and 3.6% in Test-bed 2, respectively. Figure 13c,d also shows that the AUC values for amplitude, phase and CCF are 0.9778, 0.8889 and 0.99 in Test-bed 1, and are 0.9911, 0.9649 and 0.9933 in Test-bed 2, which is an improvement of 2.1% and 11.4% using CCF compared with amplitude and phase feature in Test-bed 1 and 0.2% and 2.9% in Test-bed 2.

#### 4.2.2. SRF Improvement of the MDTW Metric

The SRF improvement is evaluated in the aspect of similarity calculation metric in this subsection. Without loss of generality, we select the proposed CCF feature to evaluate the SRF under different similarity calculation metrics. The MDTW metric proposed in this paper is compared with one TRRS metric used in [29,32] and one ED-based metric used in [28]. The similarities for CCF are calculated based on MDTW, TRRS and ED, respectively. The color maps of similarity for each metric of CCF are shown in Figure 14. Meanwhile, the ROC curves and AUC values are shown in Table 2 and Figure 15.

The color maps in Figure 14 shows that using MDTW metric can improve the SRF compared to TRRS metric and ED-based metric, since there are less grids which color is close to red in Figure 14e,f, compared with Figure 14a,c and Figure 14b,d. Figure 15a,b shows that the ROC curves of MDTW metric are closer to the TPR-axis for both test-beds, which is a good prove of the SRF improvement using MDTW. For a fixed FPR as 0.2, the TPR increases from 0.85 under TRRS metric and 0.91 under ED metric to 1 under MDTW metric in Test-bed 1, and from 0.9492 under TRRS metric and 0.9661 under ED metric to 0.9831 under MDTW metric in Test-bed 2. The results show that the MDTW metric has about 17.6% and 9.9% improvement of SRF in ROC when compared with TRRS metric and ED metric in Test-bed 1, and has about 3.6% and 1.8% improvement in Test-bed 2, respectively. Figure 15c,d also shows that the AUC values for TRRS, ED and MDTW of both test-beds are 0.903, 0.9566, 0.99 and 0.9704, 0.9661, 0.9933, which indicates that the MDTW could improve the SRF to about 9.6% and 3.5% in Test-bed 1 and 2.4% and 2.8% in Test-bed 2, compared with TRRS and ED, respectively.

It can also be concluded from Figure 14 and Figure 15 that the SRF in Test-bed 2 is better than that in Test-bed 1 because of the increase of receivers’ number, no matter what similarity calculation metric is applied.

#### 4.2.3. SRF Improvement of the CC-DTW

Since the results in the above subsections have shown that the CCF outperforms amplitude feature and phase feature in SRF and the MDTW metric outperforms TRRS metric and ED metric in SRF, we integrate the CCF and MDTW into CC-DTW, and evaluate the SRF improvement compared with one TR-based approach used in [29] and one ED-based approach used in [28]. The TR-based approach utilized the preprocessed CSI amplitude and phase information from one antenna, and used the TRRS product of amplitude and phase as the similarity calculation metric. The ED-based approach utilized the Euclidean distance between the processed amplitude and phase from three antennas as the similarity calculation metric. The gaps between the proposed method and the TR-based approach or the ED-based approach are the CSI feature processing method and the similarity calculation metric. In order to compare the SRF of these three approaches, the same training database and testing database with three antennas in the two test-beds are applied, and the CC-DTW, TR-based and ED-based approaches are performed. The SRF of these three approaches are calculated and compared for two test-beds. The results are shown in Figure 16 and Figure 17 and Table 3.

It can be concluded from Figure 16 and Figure 17 and Table 3 that the CC-DTW outperforms the TR-based approach and the ED-based approach in the experiments, and have an improvement of about 33.3% and 8.1% in ROC, and about 14.3% and 4.1% in AUC values over TR-based approach and ED-based approach in Test-bed 1, respectively. Meanwhile, in Test-bed 2, the CC-DTW outperforms the TR-based and the ED-based approach by about 0.9% and 1.2% in ROC, and about 1.5% and 1.5% in AUC, respectively. According to the results in Figure 12, Figure 14 and Figure 16, we can conclude that the improvement in SRF of the proposed CC-DTW approach is obtained by two aspects: one is the CCF processing and the other is the MDTW similarity calculation metric. It can also be noted that the performance of these three approaches are better in Test-bed 2 than in Test-bed 1, and the increase of the receivers’ number could also obtain an improvement of SRF for indoor fingerprinting.

Since the TR-based approach used in this paper as a competitor only utilized one receiving antenna in the literature, in order to compare with this approach more fairly, we also evaluate the influence of antenna number between the proposed method and the TR-based one. Experiments are conducted in Test-bed 1 with single transmitting antenna and single receiving antenna, and both the SRF of the CC-DTW and the TR-based one are calculated based on the same database. The results are shown in Figure 18. It can be concluded from Figure 18 that the CC-DTW could outperform the TR-based approach in SRF improvement. Meanwhile, comparing Figure 18 with Figure 16a,b it can be concluded that the increase of antenna’s number could also bring an improvement in SRF for both the CC-DTW and the TR-based approach.

### 4.3. Positioning Performance Evaluation and Analysis of CC-DTW

In order to prove that the improvement of SRF could bring the improvement of positioning accuracy for indoor fingerprinting systems, we conducted experiments to evaluate the positioning performance of the proposed CC-DTW method. We measured the CSI data in the two test-beds and performed the location estimation based on the “CCF_train” database, the MDTW metric and the weighted *k* nearest neighbor (WKNN) matching algorithm. We select 50 positions out of all the 79 positions in two test-beds for location estimation, 10 for Test-bed 1 and 40 for Test-bed 2, and the CSI data is collected about 100 packages at each position. The CC-DTW together with the fingerprint matching are performed for each data package. The TR-based and the ED-based approaches are also applied for the comparison, and the cumulative distribution function (CDF) of positioning errors together with the mean positioning error are shown in Figure 19.

The results in Figure 19b shows that the mean positioning error is 0.83 m, 1.17 m and 1.36 m for CC-DTW, the TR-based and the ED-based approach in Test-bed 1, and the error is 0.86 m, 1.96 m, 1.91 m in Test-bed 2, respectively. It can be concluded that the CC-DTW outperforms the TR-based and the ED-based approach in terms of positioning accuracy, and the CC-DTW is robust for the two test-beds.

### 4.4. Discussion of the Proposed Methods

In this section, experiments are conducted to evaluate the performance of the proposed method, and results are shown together with the comparison over other methods in recent literatures. We evaluated the performance of the proposed CC-DTW in two aspects: the aspect of SRF improvement and the aspect of positioning accuracy improvement.

In the aspect of SRF improvement, we evaluated the performance of the proposed CCF feature over CSI amplitude and phase features, the performance of the proposed MDTW metric over TRRS and ED-based metrics, and the performance of the proposed CC-DTW over TR-based and ED-based approaches, respectively. The CCF outperforms the CSI amplitude and phase features in SRF mainly because that the CCF integrates the information of both amplitude and phase in the formation of original CSI, which has also been proved in Section 4.2.1. The MDTW outperforms the TRRS and the ED-based metrics with the reason that MDTW metric considers not only the trend of positioning features over multiple subcarriers, but also the tolerance of measuring noise, which can be robust in indoor environments. The improvement of SRF using CC-DTW can be achieved since the CC-DTW integrates the advantages of both CCF and MDTW metric, which has been proved in Section 4.2.3.

With respect to the improvement of positioning accuracy, we evaluated the positioning performance of the proposed CC-DTW, one TR-based approach and one ED-based approach in both two test-beds. Since that the SRF is an important factor in fingerprinting systems, the improvement of SRF should also bring an improvement in positioning accuracy. The results in Section 4.3 have proved from the experimental aspect that the improvement of SRF could actually bring accuracy improvement for indoor fingerprinting system. The studies of SRF improvement in this paper could also constitute guidance when implementing a CSI-based positioning system.

## 5. Conclusions

In this paper, a fine-grained indoor Wi-Fi CSI-based fingerprinting localization method named CC-DTW is proposed, analyzed and evaluated. Since the SRF could influence the positioning accuracy of fingerprinting systems, this paper proposed two means in order to improve the SRF both in the aspect of positioning features and the aspect of similarity calculation metric. As for the positioning features, a new type of fingerprint named CCF is proposed and evaluated, which consists both the calibrated CSI amplitude and phase information. As for the similarity calculation metric, a new metric named MDTW is established and evaluated, inspired by the dynamic time warping algorithm. Meanwhile, the proposed CCF feature and the MDTW metric is integrated in the CC-DTW method to improve the SRF as well as the positioning accuracy. Experiments are conducted in two indoor office environments, and the performance of the proposed CCF, MDTW and CC-DTW are all evaluated. The improvements of SRF for all of the proposed methods are evaluated and discussed with the ROC curve and AUC indicators, and the results show that the CC-DTW outperforms the TR-based approach and the ED-based approach by about 33.3% and 8.1% in ROC curve, and 14.3% and 4.1% in AUC value under a limited bandwidth of 20MHz. The positioning performance of the CC-DTW is also evaluated through fixed-point testing, and the results indicate an improvement in positioning accuracy when compared with the TR-based and the ED-based approach. Our work could be helpful for the analysis and improvement of SRF in indoor fingerprinting systems and it could also be a guidance for performance improvement of indoor positioning systems.

## Figures and Tables

**Figure 1 sensors-19-01984-f001:**
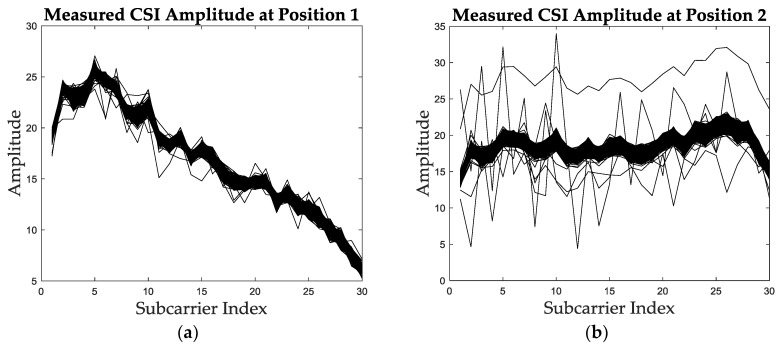
The measured CSI amplitude at different test positions. (**a**) The measured CSI amplitude at position 1. (**b**) The measured CSI amplitude at position 2.

**Figure 2 sensors-19-01984-f002:**
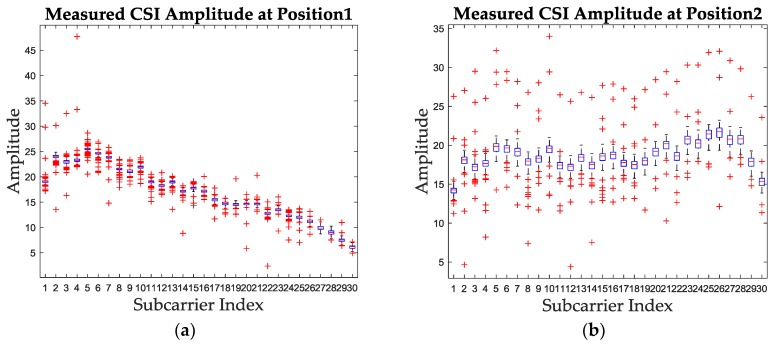
The boxplot diagram of measured CSI amplitude for two test positions. (**a**) The measured CSI amplitude boxplot at position 1. (**b**) The measured CSI amplitude boxplot at position 2.

**Figure 3 sensors-19-01984-f003:**
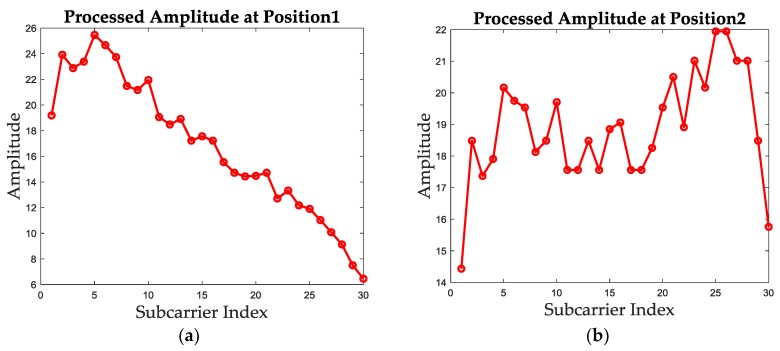
The calibrated amplitude based on mode value extraction for both test positions. (**a**) The calibrated amplitude of position 1. (**b**) The calibrated amplitude of position 2.

**Figure 4 sensors-19-01984-f004:**
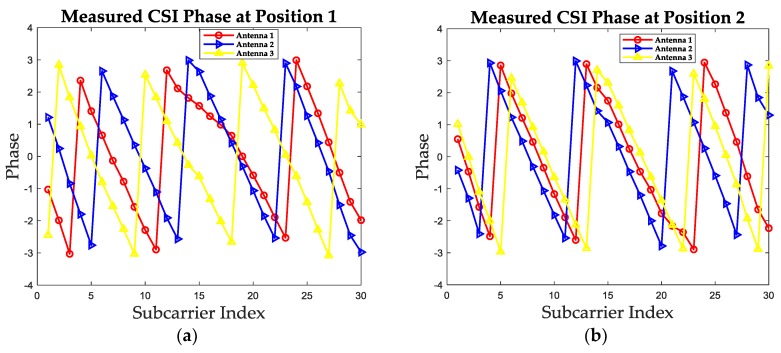
The measured phase values of three receiving antennas at two test positions. (**a**) The measured phase values at position 1. (**b**) The measured phase values at position 2.

**Figure 5 sensors-19-01984-f005:**
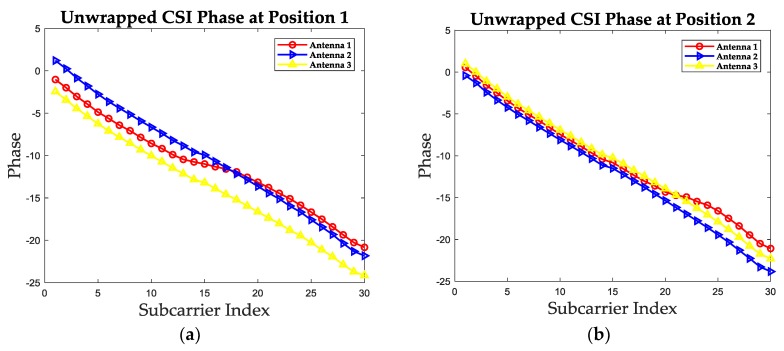
The unwrapped phase values of three receiving antennas at two test positions. (**a**) The unwrapped phase values at position 1. (**b**) The unwrapped phase values at position 2.

**Figure 6 sensors-19-01984-f006:**
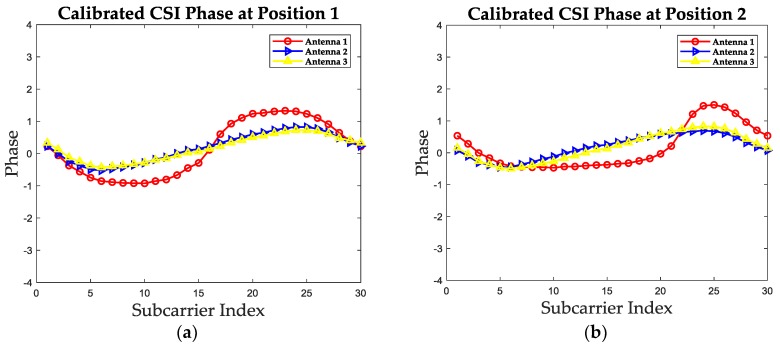
The calibrated phase values of three receiving antennas at two test positions. (**a**) The calibrated phase at position 1. (**b**) The calibrated phase at position 2.

**Figure 7 sensors-19-01984-f007:**
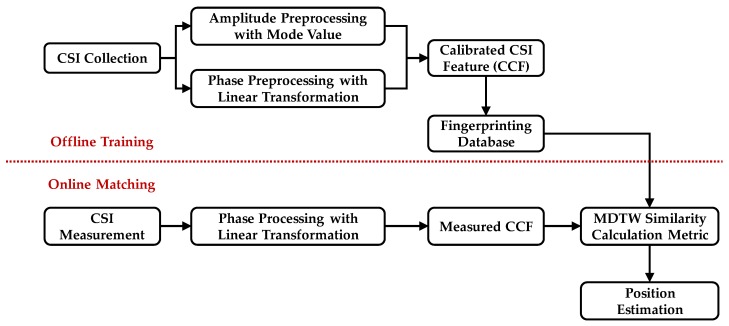
The flowchart of the CC-DTW method.

**Figure 8 sensors-19-01984-f008:**
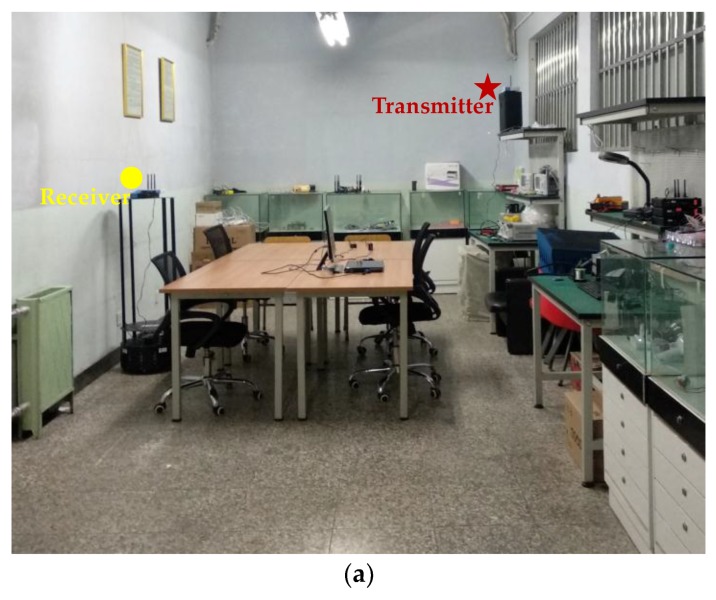
The experimental environments of Test-bed 1. (**a**) The photo of Test-bed 1. (**b**) The planer graph of Test-bed 1.

**Figure 9 sensors-19-01984-f009:**
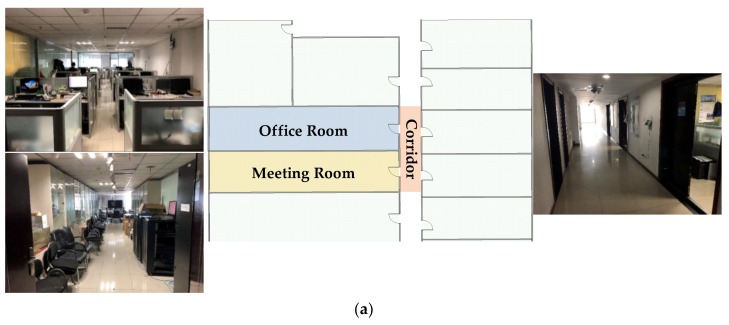
The experimental environments of Test-bed 2. (**a**) The photos of the Test-bed 2 and the receivers. (**b**) The planer graph of the Test-bed 2.

**Figure 10 sensors-19-01984-f010:**
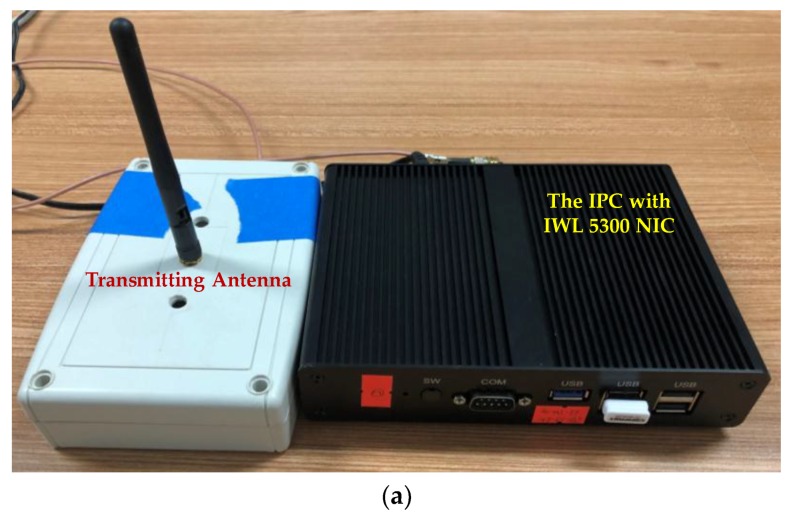
The IPC equipped with IWL 5300 NIC and antennas used in experiments. (**a**) The transmitter with one antenna. (**b**) The receiver with three antennas.

**Figure 11 sensors-19-01984-f011:**
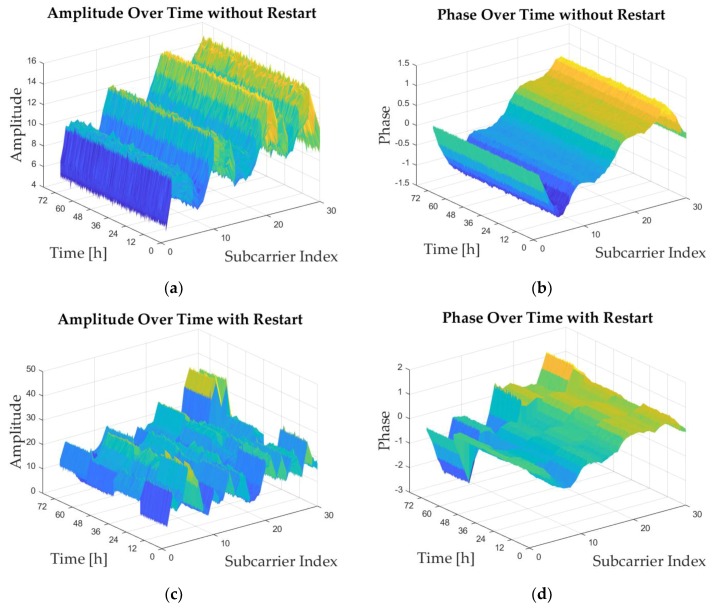
The distribution of amplitude and phase measurements over time under different transmitter conditions. (**a**) The amplitude measurements without restarts of transmitters. (**b**) The phase measurements without restarts of transmitters. (**c**) The amplitude measurements with several restarts of transmitters. (**d**) The phase measurements with several restarts of transmitters.

**Figure 12 sensors-19-01984-f012:**
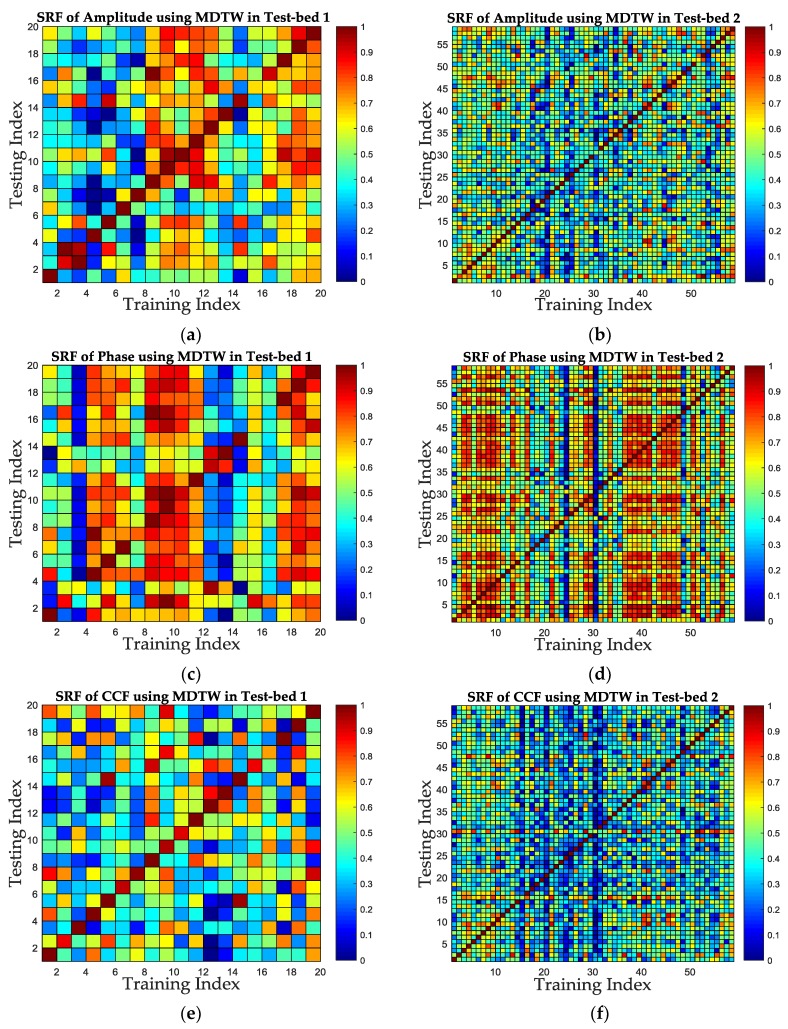
The similarity color map of three positioning features for two test-beds. (**a**) The color map of amplitude similarity in Test-bed 1. (**b**) The color map of amplitude similarity in Test-bed 2. (**c**) The color map of phase similarity in Test-bed 1. (**d**) The color map of phase similarity in Test-bed 2. (**e**) The color map of CCF similarity in Test-bed 1. (**f**) The color map of CCF similarity in Test-bed 2.

**Figure 13 sensors-19-01984-f013:**
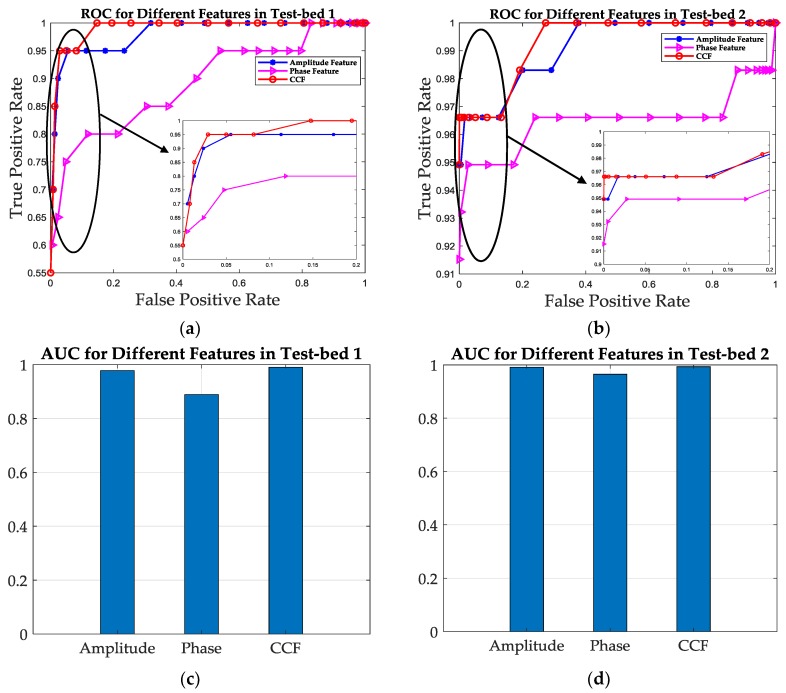
The ROC curve and AUC value comparison among three features for two test-beds. (**a**) The ROC curves for the three features in Test-bed 1. (**b**) The ROC curves for the three features in Test-bed 2. (**c**) The AUC values for the three features in Test-bed 1. (**d**) The AUC values for the three features in Test-bed 2.

**Figure 14 sensors-19-01984-f014:**
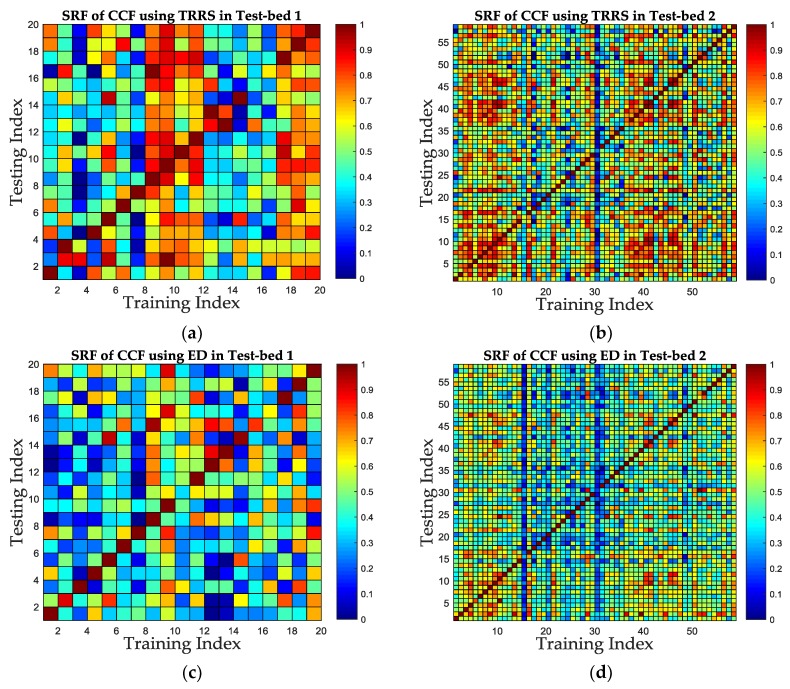
The color map of similarity under MDTW, TRRS and ED metric of CCF feature in two test-beds. (**a**) The similarity using TRRS metric in Test-bed 1. (**b**) The similarity using TRRS metric in Test-bed 2. (**c**) The similarity using ED metric in Test-bed 1. (**d**) The similarity using ED metric in Test-bed 2. (**e**) The similarity using MDTW metric in Test-bed 1. (**f**) The similarity using MDTW metric in Test-bed 2.

**Figure 15 sensors-19-01984-f015:**
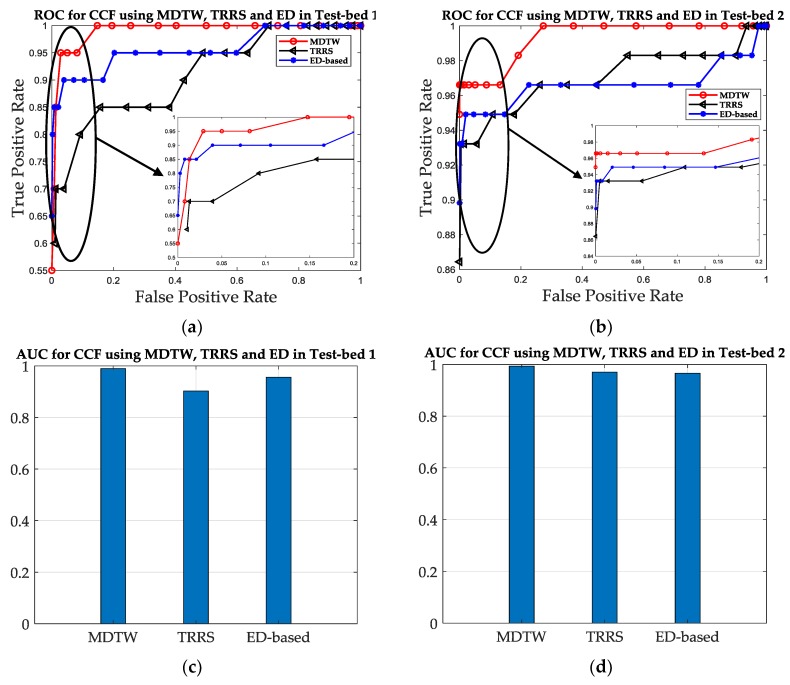
The ROC curve and AUC value comparison between three metrics. (**a**) The ROC curves for the three metrics of CCF features in Test-bed 1. (**b**) The ROC curves for the three metrics of CCF features in Test-bed 2. (**c**) The AUC values for the three metrics of CCF features in Test-bed 1. (**d**) The AUC values for the three metrics of CCF features in Test-bed 2.

**Figure 16 sensors-19-01984-f016:**
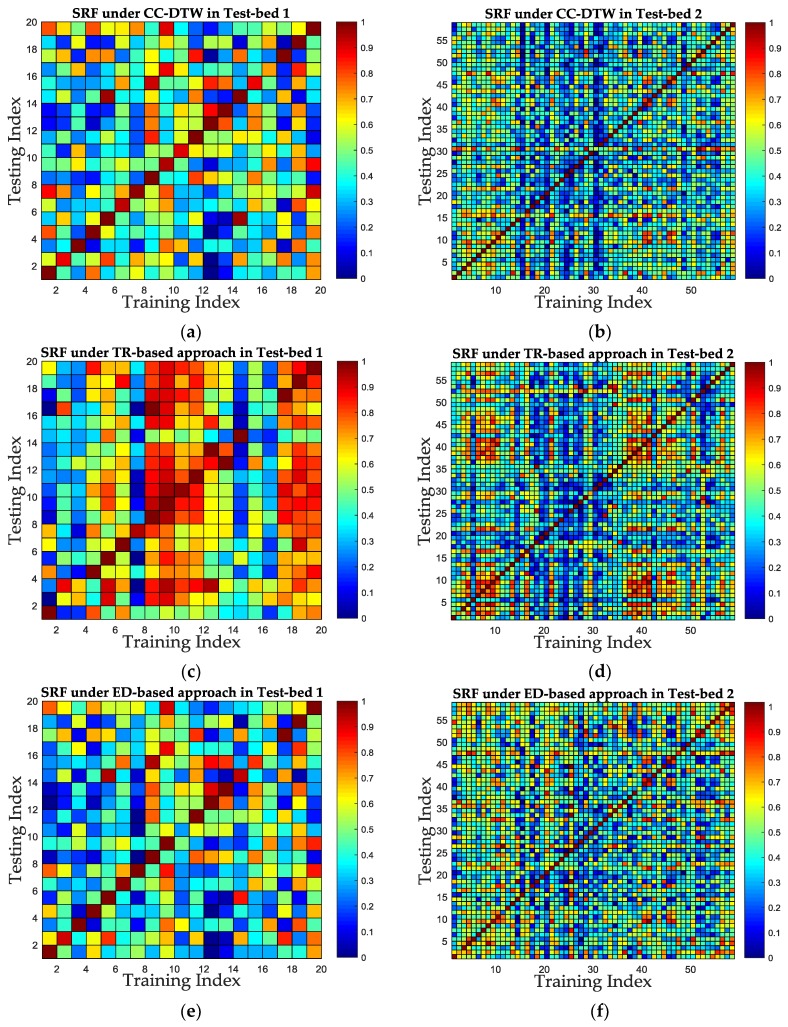
The color map of SRF using CC-DTW, the TR-based and the ED-based approach for two test-beds. (**a**) The SRF using CC-DTW in Test-bed 1. (**b**) The SRF using CC-DTW in Test-bed 2. (**c**) The SRF using TR-based approach in Test-bed 1. (**d**) The SRF using TR-based approach in Test-bed 2. (**e**) The SRF using ED-based approach in Test-bed 1. (**f**) The SRF using the ED-based approach in Test-bed 2.

**Figure 17 sensors-19-01984-f017:**
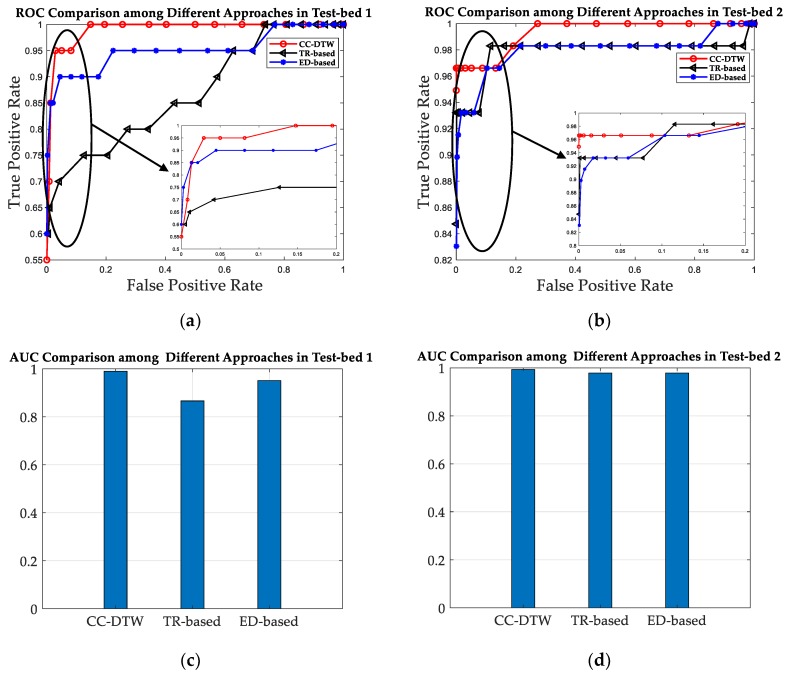
The ROC curve and AUC value comparison among CC-DTW, the TR-based and the ED-based approach for two test-beds. (**a**) The ROC curves for the three approaches in Test-bed 1. (**b**) The ROC curves for the three approaches in Test-bed 2. (**c**) The AUC values for the three approaches in Test-bed 1. (**d**) The AUC values for the three approaches in Test-bed 2.

**Figure 18 sensors-19-01984-f018:**
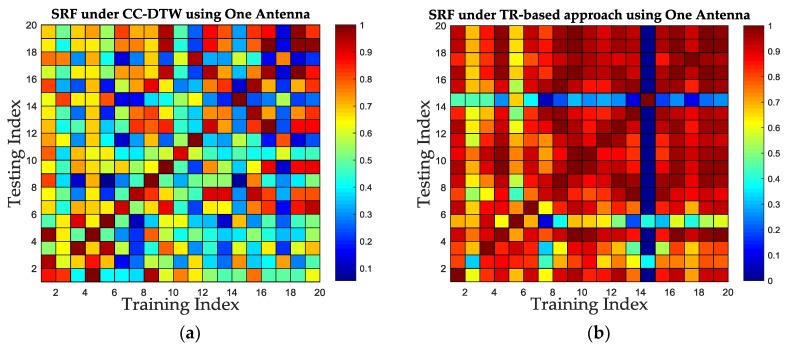
The SRF comparison between CC-DTW and TR-based approach using one antenna. (**a**) The color map of SRF using CC-DTW. (**b**) The color map of SRF using the TR-based approach.

**Figure 19 sensors-19-01984-f019:**
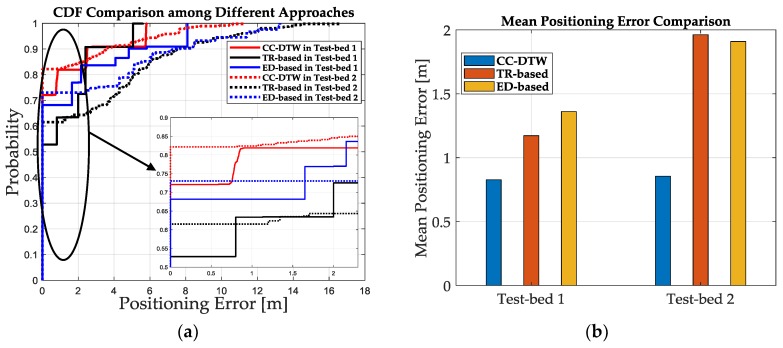
The comparison of positioning performance among CC-DTW, TR-based and ED-based approach in two test-beds. (**a**) The CDF comparison among different approaches. (**b**) The mean positioning error comparison among different approaches.

**Table 1 sensors-19-01984-t001:** The Comparison of ROC and AUC between Different Features.

	Features	Amplitude	Phase	CCF
**Test-bed 1**	**TPR (at FPR 0.2)**	0.95	0.8	1
	**AUC**	0.9778	0.8889	0.99
**Test-bed 2**	**TPR (at FPR 0.2)**	0.9746	0.9492	0.9831
	**AUC**	0.9911	0.9649	0.9933

**Table 2 sensors-19-01984-t002:** The Comparison of ROC and AUC between Different Similarity Calculation Metrics.

	Similarity Calculation Metrics	CCF with TRRS	CCF with ED	CCF with MDTW
**Test-bed 1**	**TPR (at FPR 0.2)**	0.85	0.91	1
	**AUC**	0.903	0.9566	0.99
**Test-bed 2**	**TPR (at FPR 0.2)**	0.9492	0.9661	0.9831
	**AUC**	0.9704	0.9661	0.9933

**Table 3 sensors-19-01984-t003:** The Comparison of ROC and AUC between Different Approaches.

	CSI-Based Approaches	CC-DTW	TR-Based	ED-Based
**Test-bed 1**	**TPR (at FPR 0.2)**	1	0.75	0.925
	**AUC**	0.99	0.8664	0.9509
**Test-bed 2**	**TPR (at FPR 0.2)**	0.9915	0.9831	0.98
	**AUC**	0.9933	0.9785	0.9789

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
