# Peer review of "CC-DTW: An Accurate Indoor Fingerprinting Localization Using Calibrated Channel State Information and Modified Dynamic Time Warping"

_sensors, 2019, doi:10.3390/s19091984_

Round 1
Reviewer 1 Report
I feel that the authors have addressed all my concerns in the revised version. Congratulations for the effort and I recommend acceptance of this work.
Author Response
Dear reviewer,
Thanks for your kindly comments and suggestions for our paper (Manuscript ID: sensors-479052). We also want to express our thanks for your significant and helpful comments during our revision of this paper. We replaced some figures using high-resolution ones in this revised version of our paper, in order to make the results clearer. All the changes have been highlighted utilizing green font text.
Thanks again for all of the constructive suggestions.
With our best regards,
Xiao Fu.

Reviewer 2 Report
This is a good paper to study accurate indoor localization technique.
Please add more recent related references, such as
[1] Huang, Q., Zhang, Y., Ge, Z., Lu, C. "Refining Wi-Fi based indoor localization with Li-Fi assisted model calibration in smart buildings", 16th International Conference on Computing in Civil and Building Engineering, pp. 1358-1365, 2016.
[2] Chen, J., Ou, G., Peng, A., Zheng, L., Shi, J. "An INS/Wi-Fi indoor localizaion system based on the weighted least squares," Sensors, vol. 18, no. 5, 2018.
2. The figures of 13(a)-(b), 15(a)-(b), and 17(a)-(b) are not clear with low resolution. Please replace them using high-resolution figures.
3. In section 4, please add one more section to compare and discuss this proposed work with other works in the literature. In this way, the benefits of this proposed method is highlighted.
Author Response
Dear reviewer,
Thanks for your significant comments and suggestions for our paper (Manuscript ID: sensors-479052), and we have revised our manuscript according to these helpful comments. The modified text or parts have been highlighted utilizing green font text in the revised version of our paper. The detailed point-to-point responses to these comments are attached in the following file. Thanks again for your constructive comments.
With our best regards,
Xiao Fu.

Reviewer 3 Report
The authos presente a very interesting paper about fine-grained indoor fingerprinting localisation method. The description of the methods as well as the test bed conditions are well described and the results appear to be convincing.
Author Response
Dear reviewer,
Thanks for your kindly comments and suggestions for our paper (Manuscript ID: sensors-479052). We also want to express our thanks for your significant and helpful comments during our revision of this paper. We replaced some figures using high-resolution ones in this revised version of our paper, in order to make the results clearer.
The changes of the figures are in Figure 1~6 and Figure 11~19 in the revised version. All the changes have been highlighted utilizing green font text.
Thanks again for all of the constructive suggestions.
With our best regards,
Xiao Fu.
